# Inter-Rater Reliability, Construct Validity, and Feasibility of the Modified “Which Health Approaches and Treatments Are You Using?” (WHAT) Questionnaires for Assessing the Use of Complementary Health Approaches in Pediatric Oncology

**DOI:** 10.3390/children10091500

**Published:** 2023-09-01

**Authors:** Mohammad R. Alqudimat, Karine Toupin April, Lindsay Jibb, Charles Victor, Paul C. Nathan, Jennifer Stinson

**Affiliations:** 1Lawrence S. Bloomberg Faculty of Nursing, University of Toronto, Toronto, ON M5T 1P8, Canada; 2Child Health Evaluative Sciences, Research Institute, The Hospital for Sick Children, Toronto, ON M5G 1X8, Canada; 3School of Rehabilitation Sciences and Department of Pediatrics, Children’s Hospital of Eastern Ontario Research Institute and Institut du Savoir Montfort, University of Ottawa, Ottawa, ON K1N 6N5, Canada; 4Institute of Health Policy, Management and Evaluation, Dalla Lana School of Public Health, University of Toronto, Toronto, ON M5T 3M6, Canada; 5Division of Haematology/Oncology, The Hospital for Sick Children, Toronto, ON M5G 1X8, Canada

**Keywords:** complementary health approaches, complementary and alternative medicine, questionnaire validation, reliability, validity, feasibility

## Abstract

Background: This study aimed to test the inter-rater reliability, construct validity, and feasibility of the modified “Which Health Approaches and Treatments Are You Using?” (WHAT) questionnaires in pediatric oncology; Methods: Parent–child dyads were invited to complete self- and proxy-report-modified WHAT, Pediatric Quality of Life Inventory, demographics, a diary of the child’s recent use of CHA, and a questionnaire assessing the aspects of feasibility. Parents were asked to complete a satisfaction of their children’s use of the CHA survey; Results: Twenty-four dyads completed the study. The mean weighted kappa showed strong inter-rater reliability (k = 0.77, SE = 0.056), and strong agreements between the modified WHAT and the diary (self-report [k = 0.806, SE = 0.046] and proxy-report [k = 0.894, SE = 0.057]). Significant relationships were found only between recent and non-recent CHA users in relation to the easy access to CHA (self-report [*p* = 0.02], proxy-report [*p* < 0.001]). The mean scores of the feasibility scale (out of 7.0) for the self- and proxy-report were 5.64 (SD = 0.23) and 5.81 (SD = 0.22), respectively, indicating the feasibility of the modified WHAT; Conclusions: The findings provide initial evidence of the reliability and validity of the modified WHAT and their feasibility. Further research is needed to test the theoretical relationships and further explore the validity and reliability of the modified WHAT.

## 1. Introduction

Complementary health approaches (CHA), also known as complementary and alternative medicine, are commonly used by children with cancer with percent reporting use ranging from 6% to 100% worldwide (median = 57.8%, *n* = 7219 from 34 countries) [1]. CHA encompasses a diverse group of healthcare products and practices that are not part of conventional medicine or the mainstream healthcare system, including nutritional (e.g., dietary supplements and herbs), psychological (e.g., mindfulness and spiritual practices), and physical (e.g., massage and spinal manipulation) approaches, or combinations thereof (e.g., yoga, acupuncture, and dance or art therapies) [2]. CHA is usually used in conjunction with conventional cancer treatments [1,3,4,5]. This may be beneficial to relieve the cancer symptoms and side effects of cancer therapies [6,7,8]. However, the use of some types of CHA, like herbs and dietary supplements, may present a risk of interaction with conventional cancer treatments, resulting in either greater toxicity or lower efficacy of those treatments [9,10,11,12]. Thus, failing to disclose the use of such CHA types can result in avoidable complications or even death. 

The use of CHA by children with cancer is often underreported, with disclosure rates ranging from 8% to 78% (median = 43%) [1]. Children and their parents/caregivers may hesitate to share information about CHA unless health care providers (HCPs) show acceptance and openness to discussing their use [13,14,15,16,17,18,19,20,21,22]. Fear of physician reaction [4,14,23,24,25,26], belief that HCPs lack knowledge about CHA [13,27,28], and physicians not asking [13,14,19] are common reasons for nondisclosure. It is therefore crucial that HCPs understand both the child’s use of CHA and the impacts on the child’s health and that they initiate open and non-judgmental discussions of CHA use with children and their parents/caregivers. Such discussions could be facilitated via a questionnaire-based approach [29,30], but there is a lack of thoroughly validated CHA questionnaires for this purpose [31]. To address this gap, our research team conducted a sequential three-phase research project to adapt and validate a CHA questionnaire for use by HCPs. The purpose of this questionnaire was to assess CHA use by children with cancer and to initiate clinical discussions about CHA with the children and their parents. The findings of Phases 1 and 2 offer preliminary evidence of the face and content validity of the modified electronic cancer-specific “Which Health Approaches and Treatments Are You Using?” (WHAT) questionnaires (self- and proxy-reports) in children with cancer (aged 8–18 years) [32]. This study summarizes Phase 3, which sought to test the inter-rater reliability, construct validity, and feasibility of administration of the modified WHAT questionnaires.

## 2. Materials and Methods

A prospective descriptive design was used. An adapted version of the Behavioural Model of Health Services Use [33] was used to conceptualize the underlying relationships between CHA use and the relevant evidence-based variables. This study was guided by the COnsensus-based Standards for the selection of health Measurement INstruments (COSMIN) guideline [34]. Ethical approval was obtained from the Research and Ethics Boards of the Hospital for Sick Children (SickKids; REB# 1000072728) and the University of Toronto (Protocol# 41989) before the study began. Written informed consent was obtained from patients and parents before participation. 

### 2.1. Hypotheses

The following were the study hypotheses: 

**Inter-rater reliability (H1).** 
*There will be a moderate to strong inter-rater reliability (κ ≥ 0.5) between the children’s responses to the self-report version of the modified WHAT and their parents’ responses to the proxy-report version [35,36,37].*


**Convergent construct validity (H2).** 
*There will be a moderate to strong agreement (κ ≥ 0.5) between the responses to both the self- and proxy-report modified WHAT questionnaires and the responses to related questions included in the self- and proxy-report electronic diaries of child use of CHA over the previous four weeks.*


**Hypotheses testing (H3).** 
*Based on previous research and using the adapted conceptual model (Figure 1), there are significant (α = 0.05) differences between recent and non-recent users of CHA, with recent use more closely associated with the following:*


Easier access to CHA treatments [38]. 

Greater child age [39]. 

Higher parental education [40,41,42]. 

CHA use by parents [43,44,45,46]. 

Higher socio-economic status (parent-reported of total family income and employment) [47,48,49,50].

Dissatisfaction with conventional treatment [49]. 

Longer disease duration [47].

Higher intensity of conventional treatment [1]. 

Higher Health Related Quality of Life (HRQOL) scores [50,51]. 

Positive expectations about CHA use [4,47]. 

Positive perceived health status of the person after using CHA. 

Higher satisfaction with CHA use [4]. 

**Feasibility of administration (H4).** 
*The mean scores on the five-item, seven-point Likert-type feasibility of administration scale are at least 5.0 (out of 7.0) [52], indicating that both the child self-report and parent proxy-report versions of the modified WHAT questionnaires are easy to use, understand, and follow.*


**Figure 1 children-10-01500-f001:**
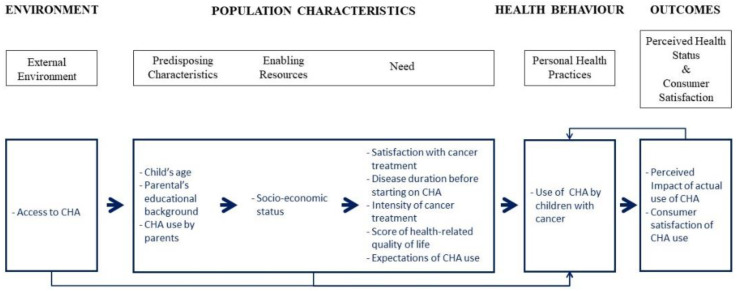
The Adapted Conceptual Model.

### 2.2. Participants 

A sample of children with cancer (8–18 years) receiving therapy at SickKids and one parent of each child was invited to participate using convenience sampling. Eligible children were English-speaking, 8–18 years of age, diagnosed with cancer, undergoing cancer treatment for at least three months after diagnosis or post-cancer treatment, and were users of at least one type of CHA according to the self- or proxy-report; the parent also needed to be English-speaking. The exclusion criteria for the child and parent participants were severe cognitive impairments or major comorbid illnesses that could preclude the questionnaires’ completion, as determined by their treating HCPs. 

### 2.3. Procedure 

Eligible children and their parents were identified by reviewing the inpatient and outpatient daily lists at SickKids, and consents were obtained from interested parent–child dyads. The participants were asked to independently complete the short weekly diary for four weeks; email prompts were sent that reminded them to complete the diary every seven days (i.e., days 1, 8, 15, and 22). At the end of four weeks, all participants received email invitations to complete short electronic surveys (both child self-report and parent proxy-report): the modified WHAT, Pediatric Quality of Life Inventory (PedsQL), the feasibility of administration survey, and demographic and health information surveys. The parents were also asked to complete a short survey on their satisfaction with their child’s CHA use. If the participants did not complete the diary or surveys within a day of receiving the relevant invitation, up to two daily reminders were sent, starting on the day following the initial invitation. If the participants failed to complete the diary or surveys by the day following the second reminder, they were considered lost to follow-up. Figure 2 shows the study schema and the assessment timeline.

### 2.4. Measures

We used the Research Electronic Data Capture (REDCap) web-based application to collect data [53]. Table 1 summarizes all measures used in this study. Since there is no gold standard measure for CHA in pediatric oncology, the research team developed brief child self- and parent proxy-report electronic diaries to record child CHA use in order to enable testing of the convergent construct validity of the modified WHAT questionnaires. The diary asked for three simple questions: the types of CHA used in the previous four weeks, the reasons for use, and whether they were helpful. The completion time of the diary was up to five minutes. 

The child self- and parent proxy-report versions of the modified WHAT questionnaires (13 and 15 items, respectively) are electronic disease-specific CHA questionnaires designed to assess and initiate clinical discussions about CHA use in pediatric oncology. The modified WHAT questionnaires have been found to have face and content validity in children with cancer [54], and they ask questions about the child’s use of CHAs since their cancer diagnosis and over the previous four weeks and about their plans for use in the future. The feasibility of administration survey comprises five items marked on a 7-point scale and has a completion time of up to two minutes [52]. This survey asks whether the modified WHAT instructions and items are easy to understand, use, and follow; whether the children and parents would take the time to complete the questionnaires (i.e., acceptability—if the child or parent are given the modified WHAT, would they complete it); and whether the completion time is appropriate [55]. A mean score of 5.0 out of a possible 7.0 would be considered a high level of feasibility [52].

The PedsQL 4.0 Generic Core Scale has shown evidence of reliability and validity and has a completion time of five minutes [56]. Cronbach’s alpha internal consistency reliability for the full-scale PedsQL has been reported as 0.88 for the child self-report and 0.93 for the parent proxy-report. Construct validity has been demonstrated using the known-groups method; PedsQL scores for healthy children were compared to those of children with cancer, and the healthy children scored higher [56]. The scale assessing parent satisfaction with CHA use by their child comprises four items marked on a 5-point scale, which takes less than a minute to complete, and asks questions about whether parents are satisfied with the outcome of the recent use of CHA and their availability, cost, and safety. There is evidence of reliability for the original questionnaire, with Cronbach’s alpha coefficients of 0.76–0.95 [57]. The research team also developed brief demographic and health information surveys to ask about sex, gender, age, race, educational background, household income, the parents’ use of CHA, the cost of CHA, the parents’ perceptions of CHA benefits, and the child’s diagnosis and cancer treatment. 

Finally, the Intensity of Treatment Rating scale (ITR) 3.0 was used to collect data from patients’ charts to categorize the intensity of the cancer treatment of the participating children based on treatment modality (surgery, chemotherapy, radiation therapy, or/and transplant) and the stage and risk level [58]. ITR 3.0 has evidence of face and content validity, and the findings of the inter-rater reliability show a high intraclass correlation coefficient of 0.86 [58].

### 2.5. Data Analysis

The data were analyzed using IBM Statistical Package for Social Sciences (SPSS) 28.0 software [59]. Descriptive statistics (means, standard deviations, and percentages) were calculated to describe the participant characteristics, and weighted kappas for the nominal variables were calculated to determine the inter-rater agreement between the responses to the child self-report and parent proxy-report WHAT questionnaires. Weighted kappas were also calculated to determine convergent construct validity between the recent CHA use reports—both self and proxy—in the WHAT questionnaires and the related reports in the diaries. The kappa levels of agreement were defined as follows: κ ≤ 0 indicates no agreement, 0.01–0.20 indicates slight agreement, 0.21–0.40 indicates fair agreement, 0.41–0.60 indicates moderate agreement, 0.61–0.80 indicates strong agreement, and 0.81–1.00 indicates almost perfect agreement [60]. Fisher’s exact tests for the nominal variables and Mann–Whitney tests for the ordinal variables were used to examine the relationship between CHA use and the related variables from our adapted conceptual model; due to the small sample size, the planned t-tests for the continuous variables could not be used, and the Mann–Whitney U tests were used instead. For all tests, the significance level was set at α = 0.05. Descriptive statistics (means, standard deviations, and percentages) were used to describe the feasibility of administration scores; since the feasibility of administration scale had no previous evidence of reliability, Cronbach’s alpha was calculated—for both self and proxy—to determine internal consistency, with a coefficient of 0.7 or higher considered acceptable [61]. 

Power and sample size were calculated to allow the identification of a significant and strong agreement (i.e., κ2 = 0.7 or greater for each item of the modified WHAT questionnaires) over and above a null value of moderate agreement (κ1 = 0.5); that is, the required sample size was calculated so that the study would be sufficiently powered such that the confidence interval around items with a strong agreement would not cover k = 0.5. With a power of 80% and alpha of 0.05, a minimum sample of 69 parent–child dyads was required [62]. Assuming 20% of the participants would drop out or be lost to follow-up [63], we aimed to enroll 83 parent–child dyads.

## 3. Results

Due to the impact of the COVID-19 pandemic on recruitment, we were unable to recruit the entire targeted sample. The results in this section are therefore preliminary. 

### 3.1. Participants 

A total of 42/106 (39.6%) parent–child dyads who were approached provided consent to participate, of whom 24/42 (57.1%) completed the study (Figure 3). The mean age of the 24 children was 14.3 years (SD = 2.8), and 50% were female, with diverse cancer diagnoses and school-grade levels. The mean age of the parents was 42.4 (SD = 8.6), and the majority were married (*n* = 19), mothers (*n* = 17), and had graduate degrees (*n* = 14). The participants’ characteristics are outlined in Table 2. 

### 3.2. Inter-Rater Reliability

Twenty-four parent–child dyads participated in determining the inter-rater reliability of the child self- and parent proxy-report versions of the modified WHAT questionnaires. Findings (Table 3) showed that 22/24 (91.7%) parent–child dyads had moderate to almost perfect agreement (κ = 0.54–1.00) between them; the other two were fair (κ = 0.232) and slight (κ = 0.087). The mean of the overall weighted kappa showed strong inter-rater reliability (k = 0.77, SE = 0.056).

### 3.3. Construct Validity

#### 3.3.1. Convergent Construct Validity 

Of the 24 children, 19 (79.2%) reported using CHA in the previous four weeks (i.e., recent use). Weighted kappa coefficients (Table 4) showed strong to almost perfect agreement (κ = 0.66–1.00) between the child self-report questionnaire and the self-report CHA diary regarding recent use for 17 (89.5%) of those 19 participants. There was no agreement (κ = 0) for the other two (10.5%) children’s responses. The mean of the overall weighted kappa indicated strong agreements between the modified WHAT and the diary (k = 0.806, SE = 0.046).

Of the 24 parents, 19 (79.2%) reported their child’s use of CHA in the previous four weeks. Weighted kappa coefficients (Table 5) showed strong to almost perfect agreement (κ = 0.74–1.00) between the proxy-report questionnaire and the proxy-report CHA diary regarding recent use for all 19 participants (100%). The mean of the overall weighted kappa indicated strong agreements between the modified WHAT and the diary (k = 0.894, SE = 0.057) 

#### 3.3.2. Relationships between Variables Measured via the Modified WHAT and Theoretically Relevant Factors 

Preliminary findings of the relationships between CHA use and the conceptually relevant variables are outlined in Table 6. As hypothesized, there was a significant difference between self-reported recent CHA users (14/18, 77.8%) and non-recent CHA users (2/6, 33.3%) regarding ease of access to CHA (*p* = 0.02); a significant difference was also found between proxy-reported CHA users (16/19, 84.2%) and non-recent users (0/5, 0%) (*p* < 0.001). Contrary to the stated hypotheses, there were no significant differences between recent CHA users and non-recent users, either self- or proxy-reported, regarding child age, parental education, the parents’ use of CHA, family income, parent employment status, the parents’ satisfaction with conventional cancer treatment, disease duration, the intensity of cancer treatment, the health-related quality of life score, the perceived helpfulness of the CHA used, the parents’ expectations of CHA use, the parents’ satisfaction with CHA use, and monthly CHA costs. There were also no significant differences, either self- or proxy-reported, between the respondents who intended a future use of CHA and those who did not regarding the perceived helpfulness of the child’s recent use of CHA, the monthly costs of CHA, and the parents’ satisfaction with the child’s recent use of CHA. 

### 3.4. Feasibility of Administration

The average scores for the feasibility of administration scale were 5.64 (SD = 0.23) for the child self-report version and 5.81 (SD = 0.22) for the parent proxy-report version (Table 7). Of the 24 parent–child dyads, most found the modified WHAT questionnaires “easy” or “very easy” to follow (16 children [67%], 14 parents [58%]), use (15 children [63%], 20 parents [83%]), and understand (12 children [50%], 13 parents [54%]); 15 children (63%) and 15 parents (63%) also found the modified WHAT to be “acceptable” or “very acceptable”.

The participants reported an average completion time of 11.54 (SD = 9.39) minutes for the self-report version of the modified WHAT and 11.92 (SD = 10.43) minutes for the proxy-report version; most of the children (16/24, 67%) and parents (15/24, 63%) found the amount of time “acceptable” or “very acceptable”. The feasibility of administration scale showed high internal consistency reliability, with Cronbach’s alphas of 0.853 for the self-report version and 0.829 for the parent-report version.

## 4. Discussion

This study evaluated the inter-rater reliability, construct validity, and feasibility of administration of the first electronic cancer-specific CHA questionnaires (i.e., self- and proxy-report modified WHAT) designed for use by HCPs to assess the children’s use of CHA and to initiate clinical discussions about CHA with the children with cancer (8–18 years) and their parents. The preliminary findings provide initial evidence of reliability and validity of the modified WHAT questionnaires and their feasibility of administration in clinical settings. The small sample size reduced the study’s statistical power and limited its ability to detect significant associations between variables. Only one hypothesized relationship was confirmed—the recent use of CHAs was significantly associated with ease of access to CHAs—but the findings offer promising initial insights into the potential of the modified electronic WHAT questionnaires. 

The modified WHAT questionnaires demonstrated initial evidence of inter-rater reliability for assessing the use of CHA in children with cancer, which was not explored previously [31]. A total of 22 out of 24 dyads showed moderate to almost perfect agreements (κ = 0.54–1.00); only two dyads showed slight (κ = 0.087) and fair (κ = 0.232) agreements. No clear reason was found to explain this low level of agreement by the two dyads. Future qualitative research involving interviews with parent–child dyads could provide insight into the reasons for such disagreements and may help to identify strategies to improve the inter-rater reliability of the questionnaires. 

Overall, the high agreement between the self- and proxy-report versions for most of the dyads suggests that the modified WHAT questionnaires capture the same information about the child’s use of CHA from the child or the parents. Self-report and proxy-report are two common methods of assessing the children’s experiences and perspectives in healthcare [64], and although the findings suggest high inter-rater reliability, it is important for HCPs to consider both when assessing the children’s use of CHA. In the context of children with cancer, self-reports may be limited by the child’s physical, emotional, and cognitive development, while proxy-reports may be biased by the proxy’s own perceptions or a lack of discussion between the parents and the child—for example, older children may use CHA without informing their parents. Thus, HCPs should seek information from both sources to gain a comprehensive understanding and should use this information to begin clinical discussions about CHA with the children and their parents.

Preliminary examination of the construct validity of the modified WHAT suggests good convergent validity, with strong to almost perfect agreement regarding CHA use between the WHAT questionnaires and both the self- and proxy-report diaries. There was no clear explanation for the disagreement (κ = 0) between the WHAT responses and the diary entries of two of the children—both female, aged 14 and 16. Further research with a larger sample is therefore needed to confirm the convergent construct validity of the modified WHAT questionnaires. 

Notably, only one of the theory-based hypotheses was confirmed, which may be due to the sample size falling short of the number calculated in the power analysis. Nevertheless, there is limited empirical data regarding the construct validity of CHA questionnaires in children with cancer [31], and the current study, despite its limitations, offers a valuable first step. Future research with larger sample sizes could adapt the current conceptual model to evaluate the theoretical relationships between CHA use and its related constructs and to more explore the potential of the modified WHAT questionnaires in clinical settings.

In this study, the electronic-modified WHAT questionnaires were shown to be feasible—short, easy to use, and not burdensome—clinical cancer-specific questionnaires for children with cancer in both self- and proxy-report versions, confirming the hypothesis of feasibility in clinical settings [65]. Questionnaires with high feasibility can improve the quality of self-reporting while minimizing missing data [64,66,67,68,69,70], and research has also shown that children and families prefer electronic screen-based questionnaires [66], suggesting that screen-based administration might help children stay focused and engaged [64]. 

The main limitation of this study is the small sample size, which may reduce the generalizability of the findings in addition to lowering the statistical power of the study, thus precluding subgroups analysis, limiting its ability to detect significant relationships between variables in hypotheses testing (H3, type II error). A larger sample could generate more accurate estimates of the reliability and validity of the modified WHAT. The study was also conducted at a single institution and included only parent–child dyads who speak English, which may limit the generalizability of the results to other contexts. The recruitment for this study was conducted over eight months at one study site. During this period, 106 eligible dyads were approached, of which only 24 dyads completed the study. The potential reasons for low enrollment and the dropout rates were that the children were overwhelmed or too sick to participate and their parents were too busy or not interested in participating in research because oncology settings are typically already research-intensive, with many calls for participants. The participants also needed to complete multiple surveys over four weeks which might impose a burden. To reduce this burden, the research team ensured that the surveys were brief and easy to complete. So, additional sites would likely be needed for future research to recruit 83 parent–child dyads at the same time. Future studies are needed to establish the reliability and validity of the modified WHAT questionnaire in other pediatric oncology settings using structural equation modeling, which may yield more precise information regarding the measurement properties. Convenience sampling is another limitation of this study as it may introduce selection bias; however, eligible participants were invited based on specific inclusion criteria, both consecutively to minimize the selection bias and over a relatively long period of time that ensured the inclusion of a diverse sample of participants.

Further research is also needed to explore pediatric oncology HCPs’ perceptions of the clinical utility of the modified WHAT questionnaires, which are important because they would shed light on the likelihood that the questionnaires will be used (i.e., acceptance) [71], their perceived clinical usefulness, and the perceived barriers and facilitators of implementation in routine clinical practice. Such perspectives could be captured using a qualitative descriptive approach [72], like the one successfully used to explore the clinical utility of psychosocial screening tools in pediatric oncology [73]. Individual semi-structured interviews with a purposive sample of oncologists and oncology fellows, nurse practitioners, and registered nurses could be conducted, which would be recorded and transcribed verbatim and then analyzed using a simple content analysis approach to generate clinical utility themes [74]. If the saturation of themes were to occur within the first 12 interviews [75], three participants from each professional group would be needed to achieve maximum variation.

Finally, future research should test the modified WHAT questionnaires with a range of cultural groups, including newly arrived immigrants to Canada. Immigrant families comprise a large and steadily growing segment of the Canadian population [76], so considering their perspectives and tailoring the modified WHAT questionnaires to their CHA use is vital, especially as CHA differ between cultural contexts [27,28,77,78,79,80,81,82]. Refinement of the questionnaires would help HCPs to ask important clinical questions and enrich their knowledge of CHA use by these important populations. As an extension, research could be conducted to test the reliability and validity of the questionnaires in low- and middle-income countries, where CHA use is high among children with cancer but disclosure rates are low compared to high-income nations [1]. The modified WHAT questionnaires could be helpful in facilitating discussions with the children and their families and in generating data to compare CHA use between such countries. To achieve this, cross-cultural validity testing and linguistic adaptations would be needed, together with low-tech alternatives for administration that would fit with the available resources.

## 5. Conclusions

This study evaluated the inter-rater reliability, construct validity, and feasibility of electronic disease-specific CHA questionnaires in children with cancer. The preliminary findings suggest that the modified WHAT questionnaires are reliable and valid for assessing a child’s use of CHAs, consistent between child and parent reports, and feasible for use in clinical settings. The pilot nature of this study and the use of an adapted conceptual model offer a valuable guide for future measurement property testing of the modified WHAT questionnaires. This study shows the potential of modified electronic WHAT questionnaires for assessing CHA use and initiating clinical discussions about CHA in pediatric oncology. Future research should address the limitations of this study, especially the sample size, and further explore the validity and clinical utility of the questionnaires.

## Figures and Tables

**Figure 2 children-10-01500-f002:**
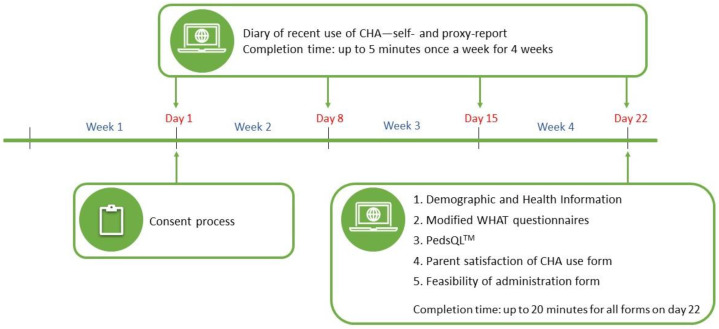
Study schema.

**Figure 3 children-10-01500-f003:**
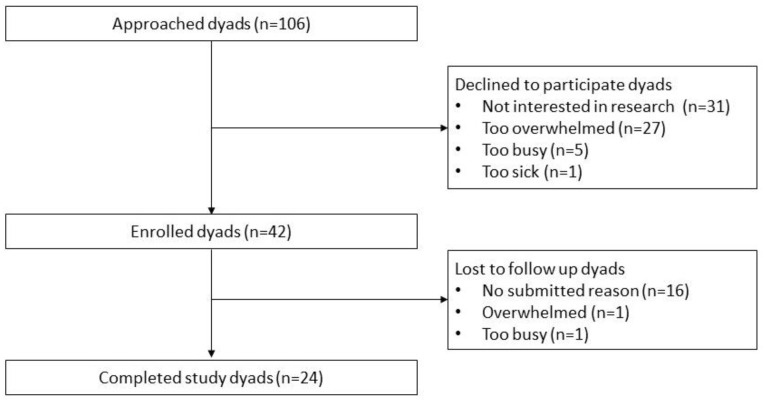
Flow diagram of dyads’ enrollment.

**Table 1 children-10-01500-t001:** Measures.

Questionnaires	Type	# of Items	Content	Time (min)	Evidence of Reliability and Validity	Hypotheses
Diary of CHA use	Self-report	3	Items: - CHA use in the past four weeks - Reasons for the use - How CHA was helpful	<5	- N/A	H2: Convergent construct validity
	Proxy-report	3		<5		
WHAT	Self-report	13	Domains: - Child’s CHA use - Variables linked to CHA use - Perceived impact of CHA use - Communication about CHA	10	- Face and content validity	H1: Inter-rater reliability H2: Convergent construct validity H3: Hypotheses testing
	Proxy-report	15		10		
Feasibility of administration	Self-report	5	Adapted items: the WHAT was - Easy to understand - Easy to use - Easy to follow - Acceptable - With a suitable completion time	2	- Internal consistency reliability	H4: Feasibility of administration
	Proxy-report	5		2		
PedsQL	Self-report	23	Domains: - Health and activities - Feeling - Getting along with others - School related issues	<5	- Internal consistency - Known group construct validity	H3: Hypotheses testing
	Proxy-report	23		<5		
Satisfaction of CHA use	Proxy-report	4	Adapted items: - Satisfaction with outcome - Satisfaction with availability - Satisfaction with cost - Satisfaction safety of CHA	<1	- Internal consistency of the original questionnaire	H3: Hypotheses testing
ITR 3.0	Chart review	4	Items: - Diagnosis - Stage or risk level - Treatment modalities - Intensity rating	N/A	- Face and content validity - Inter-rater reliability	H3: Hypotheses testing

WHAT: “Which Health Approaches And Treatments Are You Using?”; PedsQL: Pediatric Quality of Life Inventory 4.0 Generic Core Scale; ITR 3.0: Intensity of Treatment Rating scale 3.0.

**Table 2 children-10-01500-t002:** Characteristics of study participants.

**Children, *n* = 24**	
Age in years, mean (SD)	14.3 (2.8)
Female sex. *n* (%)	12 (50)
School grade level, *n* (%)	
Grade 3	1 (4.2)
Grade 4	2 (8.3)
Grade 5	1 (4.2)
Grade 6	3 (12.5)
Grade 8	1 (4.2)
Grade 9	4 (16.7)
Grade 10	4 (16.7)
Grade 11	4 (16.7)
Grade 12	2 (8.3)
Child’s cancer diagnosis, *n* (%)	
Lymphoma	8 (33.3)
Acute Lymphoblastic Leukemia	5 (20.8)
Acute myeloid leukemia	2 (8.3)
Ewing sarcoma	3 (12.5)
Osteosarcoma	2 (8.3)
Central nervous system tumour	2 (8.3)
Neuroblastoma	1 (4.5)
Other cancer diagnosis	1 (4.2)
Duration of illness in years, mean (SD)	2.8 (3.8)
Diagnosed with relapsed cancer, *n* (%)	3 (12.5)
Cancer treatment ^+^, *n* (%)	
Chemotherapy	23 (95.8)
Radiation therapy	9 (37.5)
Immunotherapy	5 (20.8)
Surgery	4 (16.7)
Stem Cell Transplant	2 (8.3)
Other treatments	2 (8.3)
Intensity of treatment score, mean (SD)	3 (0.5)
Child’s PedsQL score (self-report), mean (SD)	58.3 (15.2)
**Parents, *n* = 24**	
Age in years, mean (SD)	42.4 (8.6)
Female sex, *n* (%)	18 (75)
Parent’s relationship with the child, *n* (%)	
Biological mother	17 (70.8)
Biological father	6 (25)
Other	1 (4.2)
Parent’s marital status, *n* (%)	
Married or living common-law	19 (79.2)
Separated	2 (8.3)
Divorced	2 (8.3)
Single	1 (4.2)
Child’s caregiver at home, *n* (%)	
Child lives with one parent (mother or father)	4 (16.7)
Child lives with both parents	20 (83.3)
Ethnic background, *n* (%)	
White (Caucasian)	9 (37.5)
Black (e.g., African, Haitian, Jamaican, Somali)	8 (33.3)
Arab/West Asian (e.g., Armenian, Egyptian, Iranian, Lebanese, Moroccan)	3 (12.5)
South East Asian	2 (8.3)
Latin American	1 (4.2)
South Asian	1 (4.2)
Educational background, *n* (%)	
Graduated secondary/high school	3 (12.5)
Some college/technical school	1 (4.2)
Graduated college/technical school	3 (12.5)
Graduate degree	14 (58.3)
Prefer not to answer	3 (12.5)
Work status, *n* (%)	
Full-time	10 (41.7)
Not working	7 (29.2)
Part-time	4 (16.7)
Prefer not to answer	3 (12.5)
Spouse’s educational background, *n* (%)	
Some high school	1 (4.2)
Graduated secondary/high school	3 (12.5)
Graduated college/technical school	3 (12.5)
Some graduate school	1 (4.2)
Graduate degree	8 (33.3)
Other degree	1 (4.2)
Prefer not to answer	2 (8.3)
Missing data	5 (20.8)
Spouse’s work status, *n* (%)	
Full-time	8 (33.3)
Part-time	4 (16.7)
Not working	4 (16.7)
Prefer not to answer	3 (12.5)
Missing data	5 (20.8)
Household annual income in Canadian Dollar, *n* (%)	
Less than $24,999	2 (8.3)
$25,000 to $49,999	10 (41.7)
$50,000 to $74,999	1 (4.2)
$100,000 to $149,999	3 (12.5)
$200,000 or more	2 (8.3)
Prefer not to answer	5 (20.8)
Number of people live in the household, mean (SD)	3.8 (1.6)
CHA use by parents, *n* (%)	
Yes	11 (45.8)
CHA use intensity, *n* (%)	
High (more than five types of CHA)	2 (8.3)
Medium (2–4 types of CHA)	4 (16.7)
Low (1 type of CHA)	5 (20.8)
No use of CHA	13 (54.2)
Parent’s expectations of the child’s CHA use ^+^, *n* (%)	
CHA use would let the child feel better	17 (70.8)
CHA use would prevent the child’s illness/symptom	2 (8.3)
CHA use was safe to use	2 (8.3)
Other	2 (8.3)
CHA use would treat the child’s symptoms	1 (4.2)
Parent’s satisfaction of child’s recent CHA use outcome, *n* (%)	
Dissatisfied	4 (16.7)
Neutral	13 (54.2)
Satisfied	5 (20.8)
Very satisfied	2 (8.3)
Parent’s satisfaction of CHA cost, *n* (%)	
Very dissatisfied	1 (4.2)
Dissatisfied	8 (33.3)
Neutral	6 (25)
Satisfied	7 (28.2)
Very satisfied	1 (4.2)
Not applicable	1 (4.2)
Parent’s satisfaction of CHA availability, *n* (%)	
Dissatisfied	3 (12.5)
Neutral	12 (50)
Satisfied	8 (33.3)
Very satisfied	1 (4.2)
Parent’s satisfaction of the safety of CHA, *n* (%)	
Dissatisfied	2 (8.3)
Neutral	11 (45.8)
Satisfied	10 (41.7)
Very satisfied	1 (4.2)
Monthly cost of CHA in Canadian Dollar, mean (SD)	155 (168)
Awareness of HCPs about child’s use of CHA (parent’s report), *n* (%)	
Yes	7 (29.2)
No	7 (29.2)
Do not know	10 (41.7)
Parent’s satisfaction with conventional treatments, *n* (%)	
Very satisfied	3 (12.5)
Satisfied	15 (62.5)
Neutral	6 (25)
Child’s PedsQL score (proxy-report), mean (SD)	56.3 (15)

^+^ More than one option is possible per participant.

**Table 3 children-10-01500-t003:** Weighted kappa of the dyad’s responses on the two versions of the modified WHAT (self- and proxy-report).

Raters	Weighted Kappa (SE)	95% CI
Dyad 1	0.087 (0.089) *	−0.086 to 0.261
Dyad 2	0.756 (0.065)	0.628 to 0.884
Dyad 3	0.540 (0.094)	0.357 to 0.724
Dyad 4	0.686 (0.079)	0.532 to 0.841
Dyad 5	0.867 (0.053)	0.763 to 0.972
Dyad 6	0.987 (0.009)	0.968 to 1.005
Dyad 7	0.890 (0.037)	0.819 to 0.962
Dyad 8	0.232 (0.143) *	−0.048 to 0.512
Dyad 9	0.768 (0.061)	0.648 to 0.888
Dyad 10	0.794 (0.116)	0.567 to 1.021
Dyad 11	0.950 (0.019)	0.912 to 0.988
Dyad 12	0.835 (0.052)	0.734 to 0.936
Dyad 13	0.976 (0.016)	0.944 to 1.008
Dyad 14	0.962 (0.028)	0.907 to 1.016
Dyad 15	0.981 (0.019)	0.943 to 1.019
Dyad 16	1.000 (0.000)	1.000 to 1.000
Dyad 17	0.853 (0.053)	0.750 to 0.957
Dyad 18	0.615 (0.085)	0.448 to 0.782
Dyad 19	0.676 (0.053)	0.573 to 0.780
Dyad 20	0.749 (0.074)	0.604 to 0.893
Dyad 21	0.886 (0.082)	0.725 to 1.047
Dyad 22	1.000 (0.000)	1.000 to 1.000
Dyad 23	0.630 (0.084)	0.465 to 0.795
Dyad 24	0.763 (0.044)	0.677 to 0.848
All Dyads	0.770 (0.056)	0.660 to 0.881

* Findings did not meet the alternative hypothesis. Interpretation: for example, these findings suggest that with dyad #1, there was little to no agreement with the child’s and parents’ responses when averaged over all questions and responses.

**Table 4 children-10-01500-t004:** Weighted kappa of the children’s responses on the self-report versions of the modified WHAT and the self-report diary of recent use of CHA.

Respondents	Weighted Kappa (SE)	95% CI
Child 2	0.951 (0.037)	0.878 to 1.023
Child 3	0.000 (0.000) *	0.000 to 0.000
Child 4	1.000 (0.000)	1.000 to 1.000
Child 5	1.000 (0.000)	1.000 to 1.000
Child 6	0.746 (0.069)	0.610 to 0.881
Child 7	0.861 (0.090)	0.685 to 1.037
Child 8	0.000 (0.000) *	0.000 to 0.000
Child 9	0.865 (0.052)	0.763 to 0.967
Child 11	0.931 (0.053)	0.826 to 1.036
Child 12	0.821 (0.097)	0.632 to 1.011
Child 13	0.947 (0.050)	0.849 to 1.045
Child 14	1.000 (0.000)	1.000 to 1.000
Child 15	0.943 (0.057)	0.831 to 1.055
Child 17	0.884 (0.089)	0.709 to 1.059
Child 18	0.668 (0.133)	0.406 to 0.929
Child 19	0.770 (0.079)	0.616 to 0.924
Child 20	0.945 (0.058)	0.832 to 1.057
Child 23	1.000 (0.000)	1.000 to 1.000
Child 24	0.976 (0.018)	0.941 to 1.010
All children	0.806 (0.046)	0.715 to 0.897

* Findings did not meet the alternative hypothesis. Children 1, 10, 16, 21, and 22 reported no recent use of CHA in both modified WHAT and diary. Interpretation: For example, these findings suggest that with child #3, there was no agreement with the child’s responses for the modified WHAT and the diary when averaged over all questions and responses.

**Table 5 children-10-01500-t005:** Weighted kappa of the parents’ responses on the proxy-report versions of the modified WHAT and the proxy-report diary of child’s recent use of CHA.

Respondents	Weighted Kappa (SE)	95% CI
Parent 1	0.817 (0.101)	0.618 to 1.015
Parent 2	0.968 (0.024)	0.921 to 1.014
Parent 3	0.880 (0.060)	0.764 to 0.997
Parent 4	1.000 (0.000)	1.000 to 1.000
Parent 5	0.951 (0.049)	0.855 to 1.046
Parent 6	0.743 (0.065)	0.616 to 0.871
Parent 7	0.848 (0.100)	0.653 to 1.044
Parent 9	0.749 (0.101)	0.551 to 0.947
Parent 11	0.967 (0.025)	0.917 to 1.016
Parent 12	0.888 (0.070)	0.751 to 1.026
Parent 13	0.941 (0.055)	0.832 to 1.049
Parent 14	0.975 (0.025)	0.926 to 1.025
Parent 15	0.880 (0.086)	0.711 to 1.048
Parent 17	0.944 (0.058)	0.831 to 1.058
Parent 18	0.846 (0.094)	0.661 to 1.031
Parent 19	0.789 (0.074)	0.644 to 0.934
Parent 20	1.000 (0.000)	1.000 to 1.000
Parent 23	0.879 (0.064)	0.753 to 1.005
Parent 24	0.926 (0.033)	0.861 to 0.991
All parents	0.894 (0.057)	0.782 to 1.006

Parents 8, 10, 16, 21, and 22 reported no child’s recent use of CHA in both modified WHAT and diary. Interpretation: for example, these findings suggest that with parent #3, there was almost perfect agreement with the parent’s responses for the modified WHAT and the diary when averaged over all questions and responses.

**Table 6 children-10-01500-t006:** Preliminary findings of relationships between CHA use and its related variables on the WHAT and other theoretically relevant variables.

Recent Use of CHA	Child Self-Report (*n* = 24)				Parent Proxy-Report (*n* = 24)			
	CHA Users (*n* = 18)	CHA Non-User (*n* = 6)			CHA Users (*n* = 19)	CHA Non-User (*n* = 5)		
Variables	*n* (%)	*n* (%)		*p*-value ^	*n* (%)	*n* (%)		*p*-value ^
Easy access to CHA								
Yes	14 (77.8%)	2 (33.3%)		0.020 *	16 (84.2%)	0		<0.001 *
Parental education								
Graduated secondary/high school	2 (11.1%)	1 (16.7%)		0.373	2 (10.5%)	1 (20%)		0.452
Some college/technical school	0	1 (16.7%)			0	1 (20%)		
Graduated college/technical school	3 (16.7%)	0			3 (15.8%)	0		
Graduate degree	10 (55.6%)	4 (66.7)			11 (57.9%)	3 (60%)		
Prefer not to answer	3 (16.7%)	0			3 (15.8%)	0		
Parent’s use of CHA								
Yes	8 (44.4%)	3 (50%)		1.000	9 (81.8%)	2 (40%)		1.000
Family annual income								
Less than $24,999	2 (11.1%)	0		0.736	2 (10.5%)	0		0.656
$25,000 to $49,999	7 (38.9%)	3 (50%)			8 (42.1%)	2 (40%)		
$50,000 to $74,999	1 (5.6%)	0			1 (5.3%)	0		
$100,000 to $149,999	1 (5.6%)	2 (33.3%)			1 (5.3%)	2 (40%)		
$150,000 to $199.999	1 (5.6%)	0			1 (5.3%)			
$200,000 or more	2 (11.1%)	0			2 (10.5%)	0		
Prefer not to answer	4 (22.2%)	1 (16.7%)			4 (21.1%)	1 (20%)		
Parent’s employment status								
Full-time	6 (33.3%)	4 (66.7)		0.124	8 (42.1%)	2 (40%)		0.499
Part-time	2 (11.1%)	2 (33.3%)			2 (10.5%)	2 (40%)		
Not working	7 (38.9%)	0			6 (31.6%)	1 (20%)		
Prefer not to answer	3 (16.7%)	0			3 (15.8%)	0		
Parent’s expectation of child’s use of CHA								
Feel better	13 (72.2%)	4 (66.7%)		0.632	14 (73.7%)	3 (60%)		0.518
Prevent illness/symptoms	1 (5.6%)	1 (16.7%)			1 (5.3%)	1 (20%)		
Treat symptoms	1 (5.6%)	0			1 (5.3%)	0		
Safe to use	2 (11.1%)	0			2 (10.5%)	0		
Other	1 (5.6%)	1 (16.7%)			1 (5.3%)	1 (20%)		
Parent’s satisfaction with conventional treatment								
Very satisfied	2 (11.1%)	1 (16.7%)		1.000	3 (15.8%)	0		1.000
Satisfied	11 (61.1%)	4 (66.7%)			11 (57.9%)	4 (80%)		
Neutral	5 (27.8%)	1 (16.7%)			5 (26.3%)	1 (20%)		
	Median (Range)	Median (Range)	U	*p*-value	Median (Range)	Median (Range)	U	*p*-value
Child’s age (y)	14.9 (8.5–18)	16.2 (9.8–17.3)	30	0.109	15.3 (10.5–17.1)	11.8 (8.5–18)	27	0.145
Period since cancer diagnosis (y)	1.3 (0.3–15.8)	0.9 (0.2–10.4)	51	0.841	1.5 (0.3–15.8)	1.4 (0.4–5.3)	36	0.413
PedsQL score	60.9 (28.3–77.2)	58.7 (45.7–80.4)	47	0.640	55 (32.6–83.7)	56.5 (30.4–80.4)	32.5	0.286
Intensity of cancer treatment	3 (2–4)	3 (3–4)	52	0.873	3 (2–4)	3 (3–4)	28.5	0.107
Parent’s satisfaction of CHA use	4 (3–5)	4 (0–4.5)	48	0.710	3.1 (2.3–4)	3 (2–4)	43	0.773
CHA monthly cost (CAD)	100 (0–500)	127.5 (0–500)	42.5	0.436	100 (0–500)	50 (0–50)	36	0.427
Future use of CHA	Yes (*n* = 27)	No or not sure (*n* = 3)			Yes (*n* = 32)	No or not sure (*n* = 2)		
Variables	*n* (%)	*n* (%)		*p*-value ^	*n* (%)	*n* (%)		*p*-value ^
Helpfulness of recent use of CHA ^+^								
Helpful	14 (51.9%)	1/3 (33.3%)		0.446	14 (43.8%)	1 (50%)		1.000
Somewhat helpful	11 (40.7%	1/3 (33.3%)			16 (50%)	1 (50%)		
Not sure	2 (7.4%)	1/3 (33.3%)			2 (6.3%)	0		
Future use of CHA	Yes (*n* = 14)	No or not sure (*n* = 3)			Yes (*n* = 17)	No or not sure (*n* = 7)		
Variables	Median (Range)	Median (Range)	U	*p*-value	Median (Range)	Median (Range)	U	*p*-value
CHA monthly cost (CAD)	100 (100–500)	50 (0–500)	47	0.420	100 (0–500)	155 (0–500)	54	0.723
Parent’s satisfaction of CHA use	4 (3–5)	4 (0–4.5)	42	0.260	3 (2–4)	3 (2.3–5)	57	0.872

^ Fisher–Freeman–Halton Exact Test. ^+^ More than one option is possible per participant. * Denotes statistical significance *p*-value < 0.05; all other *p*-values were not significant.

**Table 7 children-10-01500-t007:** Feasibility of administration testing.

Administration Feasibility Questions	Child’s Self-Report *n* = 24		Parents’ Proxy-Report *n* = 24	
	Mean (SD)	Cronbach’s Alpha	Mean (SD)	Cronbach’s Alpha
Were the instructions and questions in the WHAT questionnaire easy to understand? *	5.54 (1.29)	0.853	5.67 (1.37)	0.829
Was the WHAT questionnaire easy to use? *	5.67 (1.72)		5.83 (1.69)	
If you were given the WHAT questionnaire, would you complete it? *	5.71 (1.06)		5.83 (1.05)	
Was the layout of the WHAT questionnaire easy to follow? *	5.63 (1.52)		5.83 (1.38)	
How would you rate the amount of time taken to complete the WHAT questionnaire? **	5.67 (1.40)		5.88 (1.36)	
Overall	5.64 (0.23)		5.81 (0.22)	

* Participants were asked to rate each question using 1–7 scale, with 1 being very unlikely and 7 very likely. ** Participants were asked to rate each question using a 1–7 scale, with 1 being too long and 7 too short.

## Data Availability

No new data were created or analyzed in this study. Data sharing is not applicable to this article.

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
