# Peer review of "Inter-Rater Reliability, Construct Validity, and Feasibility of the Modified “Which Health Approaches and Treatments Are You Using?” (WHAT) Questionnaires for Assessing the Use of Complementary Health Approaches in Pediatric Oncology"

_children, 2023, doi:10.3390/children10091500_

Round 1

Reviewer 1 Report

The paper has no major flaws but the number of patient-parent dyads is very low precluding any analysis of subgroups and any more interesting conclusions.

It would be fair for the authors to present the number of "dyads" approached and surveyed versus the capacity of the centre over the period?

What about the approch to complicated or time-consuming and thus such low capture?

Please discuss if the approach preformes better that wht is already published in (e.g. surveys)?

Reviewer 2 Report

Little is known about children undergoing oncological treatments.  The use of the adaptive conceptual framework fora the investigation of psychometric properties of an assessment instrument for pediatric patients is novel and of great importance to demonstrate the validity and reliability of the evaluation instrument.  Although this is a pilot study with a small sample size of children and their parents, the research paper presents preliminary results on the interrater reliability and construct validity.  It is important to document the need for conducting more rigorous research on the validity and reliability of the treatment outcome instrument.  For instance, the use of structural equation modeling and related psychometric assessments may yield more precise information regarding the psychometric properties of the measurement instrument.  In other words, the authors should elaborate how future studies should be conducted.
